# Qualitative Study of the Implementation and Potential of a Social Intervention at Nursing Homes in Denmark

**DOI:** 10.3390/ijerph18041808

**Published:** 2021-02-12

**Authors:** Anne Sophie Mikkelsen, Maria Kristiansen

**Affiliations:** Center for Healthy Aging, Section for Health Services Research, Department of Public Health, University of Copenhagen, 1014 Copenhagen, Denmark; makk@sund.ku.dk

**Keywords:** social relations, nursing homes, narratives, implementation, qualitative analysis

## Abstract

(1) Background: The effect of social relations on health and wellbeing is well documented. However, knowledge about social interventions specifically in nursing homes and their potential for health and wellbeing is inadequate. In this qualitative study, we explore the implementation of a social intervention entitled Tell Stories for Life implemented in Danish nursing homes. (2) Methods: Through a qualitative multi-perspective longitudinal approach, nursing home residents and employees were interviewed from May–December 2016 (N = 14). The authors made participatory observations and took field notes. (3) Results: The intervention did not appear to establish or strengthen social relations between nursing home residents. However, nursing home residents enjoyed participating, narrating and having someone listen to their stories. The identity of nursing home residents and their relationships to nursing home employees facilitating the intervention appeared to be strengthened. Barriers were related to lack of support from management, nursing home employees’ educational backgrounds and experiences, and nursing home residents’ cognitive ability. (4) Conclusions: This study found that the Tell Stories for Life intervention did not appear to strengthen and establish social relations among nursing home residents. However, we found that there might be potential for strengthening residents’ sense of identity and the relation between residents and nursing home employees.

## 1. Introduction

Population ageing poses challenges and opportunities for health and social care delivery worldwide [1,2,3]. Social relations are, together with their negative manifestations such as loneliness and social isolation, part of the complex set of factors shaping health in old age [4,5]. Studies have shown that people with weak or straining social relations have higher morbidity and mortality as well as lower levels of physical and cognitive functional levels than people with strong social relations [6,7,8,9].

In recent years, the proportion of older people living in nursing homes has risen, and those moving into nursing homes tend to be in poorer health, have a lower functional and cognitive abilities, and a weaker social network than older people in general [10,11,12,13]. Moreover, nursing home residents have often experienced stresses such as loss of a spouse, relatives or friends [14,15]. In Denmark, nursing homes are publicly run institutions housing older people in individual units, with hired staff to provide care for the residents 24 h a day [16]. The local municipality refers an individual to a nursing home based on professional assessments of the individual’s physical, mental and social functioning and need for care [16].

In a public health perspective, it is important to understand how our knowledge about the effect that social relations have on our health might translate into meaningful and effective preventive interventions. Despite the vast amount of research in the area of social relations and health, we still seem to be on more uncertain ground when it comes to how to act on this knowledge in terms of preventive measures. Numerous original research papers and a range of systematic reviews have to date reported on interventions related to aspects of social relationships such as perceived social support, social isolation, social capital, or loneliness among adults [7,17,18,19,20,21,22]. Given the well-documented health impact of social relations and related aspects of social relationships among adults, and given the growing ageing population, we argue that it is pertinent to identify effective interventions suitable to this particular population group.

Boundaries between different typologies of interventions are blurry. In this study, we understand social interventions as interventions, which are centered around human social interactions in groups of individuals and which seek to address social relations. A systematic literature review by Mikkelsen et al. identified a limited number of studies exploring the effects of social interventions among older people living in nursing homes [23]. Findings in the systematic review indicate that various types of social intervention may have potential in relation to a range of different outcome measures such as depression, wellbeing, life satisfaction, loneliness, cognitive performance, identity, quality of life, social engagement and self-transcendence [15,24,25,26,27,28,29,30,31,32]. Another systematic review by Franck et al. focusing on interventions addressing social isolation and depression among aged care clients identified one reminiscence therapy intervention, which had a positive impact on social isolation and depression [33], and two interventions with a positive impact on depression alone. One intervention focused on establishing compensatory strategies through Wii technology [34] whereas the other addressed the effectiveness of a gardening programme at nursing homes [35]. In support of conclusions from a number of previous review studies [7,17,22], Franck et al. point to specific intervention components which may be significant for successful interventions: high-quality training and continuous support of intervention facilitators, involving the participating older people in the planning, implementation and evaluation of the intervention, and utilising existing community resources [21]. Moreover, two previous review studies point specifically to group-based or social support intervention as most effective, especially when theoretically based [7,22].

Generally, interventions addressing social aspects of individuals’ lives such as their relations with other people and their experience of available social support are complex since what such interventions aim to address and change might rely on embedded and habitual behaviours of participants and the context in which the intervention is implemented [36,37]. It is important to acknowledge issues of complexity, context and the interplay of intervention components as these factors come together in shaping the implementation process if we want to generate knowledge about what works for whom and under what circumstances [38,39,40].

With this study, therefore, we wish to explore the implementation of social interventions in a nursing home setting. The narrative intervention *Tell Stories for Life* implemented in two specific Danish nursing homes served as the empirical field of research and will be described in the following.

### Tell Stories for Life: A Narrative Life-Story Intervention

The Tell Stories for Life intervention was developed by the Danish private non-profit EGV (Social Inclusion of Older Adults) Foundation [41]. The intervention’s theoretical foundation is based on narrative therapy, which sees potential for change in the process of sharing significant life stories, and focuses on social relations not merely as present or established during the intervention activities, but rather as lasting and sustainable beyond the intervention period. The rationale of the intervention is that sharing life stories in small supervised and facilitated groups creates a space where links to personal biographies are established and maintained. Through storytelling, older adults may thus establish a foundation for strengthening and forming social relations that may prevent or reduce feelings of loneliness. As the EGV Foundation states in their description of the intervention: ‘Through stories and social context, the participants connect, they become new witnesses to each other’s life stories and they establish a new outlet for social relations’ [41]. Practically, the *Tell Stories for Life* intervention was carried out at nursing homes with nursing home employees as group leaders facilitating the intervention and nursing home residents as participants in the intervention. The core intervention activity involved facilitating and supervising group sessions organised around participants sharing their life stories in small groups of three to five people. The nursing home employees facilitating and supervising the group sessions had undergone a three-day training course organised by the EGV Foundation on how to recruit participants, facilitate the group sessions, supervise and follow-up, if needed. The joint implementation by the EGV Foundation and Copenhagen Municipality began in spring 2016 and continued throughout 2017, and each of the intervention groups met for eight to ten weeks. From time to time group leaders and participants together decided on a theme for each session to centre the story sharing around. For example, it could be travels, cities that they would like to see, music or movies that made an impression. The nursing home employees were to facilitate the sessions so that all participants would get the possibility to talk and share any story they might find meaningful and to ensure that while telling a story they would not be interrupted by the other participants.

## 2. Materials and Methods

We used a multi-perspective longitudinal approach in which intervention participants and group leaders facilitating the intervention were interviewed at the beginning of the intervention and again after the intervention. Participatory observations were carried out and field notes taken throughout the intervention period.

### 2.1. Setting and Recruitment

The selection of implementation settings (nursing homes) and informants at the chosen settings (residents participating in the intervention, and nursing home employees facilitating the intervention) were dependent on gaining access through different gatekeepers. First, implementation settings were chosen in collaboration with the implementing partners, namely the EGV Foundation and the Municipality of Copenhagen. Two nursing homes in Copenhagen were selected for the study. Then, the first author contacted the three nursing home employees at the two settings who all agreed to participate in the study. Two of the nursing home employees worked at the same nursing home and facilitated an intervention group together. As part of their group leader training, the employees at the nursing homes were responsible for selecting and inviting participants for their intervention groups. Recruitment of intervention participants for this study was done through the nursing home employees who asked all the participants in their intervention groups to be interviewed for this study. Finally, from each of the two groups two out of five participants agreed to be interviewed. Hence, in total three nursing home employees and four nursing home participants were interviewed. Additionally, from the first initial step of setting up the contact with implementing partners and being invited into this field of practice, we had several informal and non-recorded talks with representatives from the implementing partners. These initial encounters, which we at first merely considered as the formal preparations, have served as invaluable accounts, establishing an understanding of the complexity of the context of implementation. This way, we held additional informal and non-recorded meetings with representatives from each of the partner organisations. Doing this, we focused on mutual sharing of views on the implementation and potential of the intervention, and we shared our preliminary perspectives based on interviews and observations.

### 2.2. Semi-Structured Interviews and Participant Observations

At the three-day training course, in which nursing home employees were trained in the intervention methodology, the first author and researcher presented herself, background and aim of the study, and made initial field observations. She made further field observations during the group sessions and carried out interviews with intervention participants and nursing home employees in private rooms at the nursing homes. All participants received an information folder about the study including contact information on the first author conducting the interviews and field observations. She carried out interviews in a semi-structured manner using pre-developed flexible interview guides: developed separately for participants and nursing home employees respectively. Interviews varied in length from app. 30 to 45 min. Field observations were guided by a pre-developed observation guide [42]. The interview guide for the intervention participants entailed questions to uncover aspects of their perceived social life and social relations at the nursing home, their motivation for participating in the intervention and their experiences with it. The interview guide for the nursing home employees included questions to uncover their experiences with facilitating the intervention, lessons learned and perceptions of how relevant and successful the intervention had been at the nursing home. An overview of research questions and the related main- and sub questions used in the interview guides is given in Appendix A and Appendix B

### 2.3. Analytical View and Theoretical Approach

In order to analyse the implementation of the social intervention, we applied the normalization process theory (NPT) developed by Carl May [43]. Interventions in healthcare and social settings engage multiple stakeholders, rely on often embedded and habitual behaviours, and, as noted above, are often implemented in complex contexts and enrol participants with varying characteristics [36,37]. The NPT helped us to understand how intervention participants and nursing home employees involved in the implementation of the Tell Stories for Life intervention invested in contributions that mobilised the capabilities of the different inherent components of the intervention. Furthermore, the theory acknowledges that contributions from participants do not happen in a vacuum but rather draw on structural and cognitive resources, which then also become relevant to explore. In the analyses, we focused on selected components of the NPT that emerged in the material; the capabilities (qualities of workability and integration into practice) inherent in the intervention, the capacity (rules and roles that govern behaviour around the complex intervention) to cooperate and coordinate action amongst practitioners, and the potential (the ability to translate capacity into action) [36,37,43].

### 2.4. Data Analysis

The first author recorded and transcribed the interviews and observation notes, and both authors subsequently organised, coded and analysed the interviews in Nvivo. When transcribing the interviews the first author recorded words, ignoring pauses and sounds. Both authors did the thematic coding and analyses guided by predefined components described in the NPT, while at the same time adhering to an ongoing iterative manner allowing ourselves to make changes and adjustments in the structure and themes identified under each of the included theoretical components (capabilities, capacity and potential).

### 2.5. Ethical Considerations

The study, including collection of data and data processing, was conducted in accordance with the ethical principles for medical research as set out in the Helsinki Declaration [44]. Due to the small scale design in which relatively few individuals, mostly from the same local area, were interviewed, securing anonymity of the respondents was particularly important. We obtained written informed consent from all participants, presented findings in anonymous form and took care to ensure that individuals were not identifiable. In order to secure respondents’ anonymity names are replaced with a capital letter in the table and in the subsequent sections. Measures were taken to ensure appropriate reflexivity during the planning and data collection process, and throughout the analysis and dissemination phases. Preliminary findings were presented and discussed with partner representatives—the EGV Foundation and Municipality of Copenhagen—and a multi-disciplinary research group at the University of Copenhagen in order to gain additional perspectives on our analyses.

## 3. Results

In this section, we will present findings from analyses of the implementation process as perceived by intervention participants and nursing home employees respectively. These findings will subsequently guide the discussion of the potential of the intervention. Overall, we observed no substantial developments over time when analysing the pre- and post-intervention interviews and therefore, we will instead focus on the different perspectives of the implementation of the *Tell Stories for Life* intervention as expressed by the intervention participants and the nursing home employees.

In Table 1, we present an overview of the characteristics of the study participants: four intervention participants, and three employees at the two included nursing homes, these employees serving as intervention group leaders.

### 3.1. Capabilities to Enact the Tell Stories for Life Intervention

According to the NPT, the capabilities of intervention participants and nursing home employees to enact the intervention depend on how the intervention is used which relates to its workability as well as the confidence in its use i.e., its integration.

After the intervention period, findings indicated that none of the participants had established new social relations or strengthened existing ones with other participants. Moreover, participants expressed no wish to continue meeting in the intervention groups once these were no longer facilitated by nursing home employees. Overall, there seemed to be little confidence in the intervention models’ sustainability as intended by the implementing partners— i.e., that the groups should continue to meet of their own accord after the facilitated group sessions. When asked during the interviews, neither the nursing home employees nor the nursing home residents believed that the intervention groups would be able to continue without someone to facilitate the sharing of life stories.

Although the intervention did not appear to establish or strengthen social relations among intervention participants as intended we found indications of other positive experiences with the intervention. Generally, participants stated that they liked the *Tell Stories for Life* intervention, saying they would participate again if invited. More specifically, one participant expressed particular enthusiasm regarding the intervention, explaining during both her interviews how she enjoyed narrating and being listened to—thereby reliving past memories—and passing on positive memories to other participants:


*E: Well, yes. When you have had a great time experiencing these stories, then it’s of course lovely to pass them on to others. It’s like reliving them again. And then if people at the same time are interested, then it’s fun. It’s kind of like, they are tasting some of it again.*
(Participant E)

She described how listening to other participants’ stories triggered yet more memories from her own past:


*E: … You inspire each other. Someone tells something. then you can suddenly say, ‘Ah yes, and there’s this and there’s this’. But really, the experience was that I was never as preoccupied with their stories or travels.*
(Participant E)

Whereas nursing home residents expressed little confidence in the intervention in terms of its effect on social relations and sustainability, nursing home employees expressed a more positive view. They expressed a confidence which was based on a perceived need among the older residents to tell stories and be listened to, and on an observed strengthened sense of identity among the intervention participants. Also, the nursing home employees expressed positive experiences with the intervention in terms of fulfilling a need to tell one’s story and be listened to, thereby strengthening a sense of identity as someone else than an older person living at a nursing home.

Observing one of the group sessions at one of the nursing homes, we saw that while waiting for the nursing home employee to start the session, the participants seemed unengaged and unaware about what were about to start. The common theme for the session was ”music and memories”, and as the nursing home employee put on the first song, the participants recognised it and it initiated conversations and stories about when they were all young and dancing. The nursing home employee contributed with a personal story about when she was attending standard dance, and it appeared that one of the nursing home residents met his wife when dancing. Another participant noted that as children in a period of wartime, there was a lot of dancing. Albeit being cognitively impaired, and not seemingly engaged in this particular intervention, the setting and the common theme thereby appeared to established a common ground and understanding among the participants and the nursing home employee facilitating the intervention.

Another example of how the common themes contributed to establish a common ground for the participants albeit being cognitively impaired, were when two participants discovered that they shared the same profession and passion as mechanics—one for cars and the other for plains.

In terms of potential for integrating the intervention into action, two of the nursing home employees expressed confidence in the intervention. They talked about seeing a need for residents to tell their stories and listen to others, observing a strengthening of identity and meaning in life among some of the participants. One of the nursing home employees expressed how, in her view, the residents at the nursing home had a certain identity as an older person living at a nursing home and that by sharing stories from before their nursing home identity, they felt revitalized:


*T: This, I think, is the best example of why it makes really a lot of sense to have these kinds of groups here. Maybe especially at nursing homes, where you—I mean your identity is affected by the fact that now you are the kind of person who lives at a nursing home, and now you are the kind of person who is sick and needs help. So maybe it makes even more sense to hold on to these big things in life that you did and which were good or special.*
(Nursing home employee T)

The same nursing home employee described how the intervention seemed to train the participants’ rather ‘rusty’ talking skills, so although as she put it, the participants might not have established new relations and made friends, by having this sort of talking group, they revived forgotten or rusty skills useful for other social encounters.

However, another nursing home employee was less enthusiastic—not considering the *Tell Stories for Life* intervention suitable at a nursing home, nor seeing a need for it:


*M: It takes time. That you know from yourself. You kind of need half a bottle of red wine before you start sharing what’s in your heart or mind, right? And then, when you sit for an hour with people that you’re not used to sharing with and opening up to, then it takes something extra. And they were getting there but I think it takes a dedicated and permanent employee, who believes in the project, who needs to carry it and really hold on to it.*
(Nursing home employee M)

Moreover, with this statement, the nursing home employee drew our attention to the challenges there might be related to sharing private and intimate stories with people with whom there is not already an established relation. For many people—regardless of age, social background etc.—it takes time to establish a relation and maybe even become friends or at least feel safe enough in order to open up and share personal stories and thoughts.

A need to adapt the intervention to the specificities of the local implementation context as well as to the preferences of those implementing the intervention (here the nursing home employees) emerged through the course of this study. During interviews and in the field observations, we found examples of how the intervention was adapted to fit the implementation context. In particular, based on our observations we saw adaptation in terms of how the group sessions were facilitated, such as the themes chosen, and the nursing home employees’ ways of guiding and facilitating the talks and interactions during the group sessions. For instance, some of the nursing home employees helped the more cognitively weak participants to speak in the group sessions by asking them more in-depth questions, or reminding them what stories they had told earlier—otherwise, as one of the nursing home employees pointed out, the participants might easily have lost the thread, or stopped, or refused to say anything. In doing so, the nursing home employees carefully considered the cognitive state of the individual participants and reflected on how to enable them to take an active part in the conversations. This may indicate the intervention’s workability (how it is being used) and that the intervention model is viewed as rather flexible, so that nursing home employees might diverge from running it according to the described model. The individual nursing home employees thus maneuvered between the task of implementing this specific intervention model and the structural realities within which they implemented it. One of the nursing home employees who had worked at the nursing home for three years described how she assisted the intervention participants more in telling their stories than initially intended by the intervention model and why she did it:


*ED: But the project itself, I find a bit difficult because I try to work in a different way... Here one [person] at a time tells an entire story, and then you need to raise your hand and get the permission to ask a question. That I think is a bit rigid. … When we sit here in the group, … everyone should have the possibility of telling their story. And I can see how they light up… when they … move into their memories and tell their stories. Then their faces light up … and they think it is nice to be allowed to talk about things they have experienced in their life.*
(Nursing home employee ED)

### 3.2. Capacity and Potential to Enact the Tell Stores for Life Intervention

According to the NPT, the capacity to enact the intervention depends on structural resources such as the cultural norms and values of the specific institutional or organizational setting, and the potential to translate capacity into action depending on individuals’ cognitive resources.

One of the nursing home employees expressed in both of her interviews as well as during informal talks before the group sessions how she was struggling with prioritising the time needed for implementing the *Tell Stories for Life* intervention. She described how initially when introduced to the intervention she was promised extra time from her manager, which in the end did not materialise. Consequently, she had difficulties prioritising the time to implement the Tell Stories for Life intervention while simultaneously carrying out her core caring tasks at the nursing home. Related to the issue of time and having to prioritise the intervention, all the nursing home employees expressed how there were also many other competing activities at the nursing home, hence a difficulty in prioritisation. Consequently, it was sometimes hard for the participating residents to distinguish this particular intervention from other activities:


*ED: …and then we have babies, and dogs and we have the cycling and then we have other activities... and we have just gotten two new volunteers and other volunteers that I also need to get started here—one who would like to read for the residents and so on and so on… so there’s a lot of activities, which is why I haven’t started up a new round of Tell Stories for Life yet.*
(Nursing home employee ED)

Despite the limited number of participants, we found indications that the nursing home employees’ educational background and previous experience shaped the engagement with intervention:


*ED: Well, I am a trained social worker, right? So I have some experience in how to gather a group, and get the different members of it to step forward and talk and so on. That experience I’ve used a lot.*
(Nursing home employee ED)

Moreover, adding to the nursing home employees’ perspective, representatives from the partnering organisation explained how generally, where care personnel implemented the intervention, the process of implementation was less successful than where an activity worker, volunteer or other type of employee implemented the intervention. As one of the nursing home employees expressed directly:


*ED: It makes it a lot easier, because if you were also a care-giver and had to prioritize it [the intervention] when for instance colleagues were sick, so that your colleagues would have to work even faster to get through the day, then I think people would start to comment negatively. But they don’t because it’s us who implement it [and an activity worker and dietician]. We’re dedicated to these kinds of task.*
(Nursing home employee ED)

## 4. Discussion

### 4.1. Potential of the Intervention

In this qualitative study, we explored the implementation and potential of a social intervention entitled *Tell Stories for Life* in a nursing home context. Overall, our results indicated that the narrative intervention did not establish or strengthen social relations between nursing home residents participating in the intervention. As described in the results section, both the nursing home employees and residents participating in the intervention expressed little confidence in the intervention’s sustainability—particularly in terms of participants being too physically limited (e.g., needing assistance to move from their apartment within the nursing home to the intervention site) and cognitively limited (e.g., difficulties remembering to attend the group sessions). Hence, the nursing home residents would not be able themselves to gather in the groups and maintain a conversation without a nursing home employee to facilitate the process. We saw that nursing home residents in general had difficulties engaging in meaningful conversations with us during the interviews, and that we had to remind them of the intervention—sometimes several times during the interview. During the interviews with the intervention participants, we sought to introduce ourselves and have the participant ‘tune in’ on the *Tell Stories for Life* intervention. We did this the first time we met the participant as well as the second time, which was a short while after the intervention had ended. Still we found that some of the participants had difficulties focusing on and remembering the intervention. Initially, this led us to question the suitability of nursing home residents to participate in an intervention using narratives. As opposed to the earlier-cited argument about the importance of activities to establish a sense of purpose among nursing home residents [11], this might indicate that the mere presence of facilitated social activities is not in itself enough for residents to engage with other people.

Moreover, when addressing the structural resources related to the implementation of the *Tell Stories for Life* intervention, we found that lack of management support regarding allocation of extra time to facilitate the intervention were a clear barrier to the success of the implementation—especially if the nursing home employee also had to perform nursing tasks. In relation to this, those nursing home employees employed as non-caring personnel (e.g., as dedicated activity workers), expressed more freedom and possibilities to prioritise this particular intervention in their daily work. Yet, when interviewed after the intervention period, they explained they had not had time to start up a new intervention group, owing in part to too many other activities. The described struggle with prioritising the time to implement the intervention seemed to have resulted in an ambiguity towards the intervention where the nursing home employee questioned the need and relevance for this particular intervention while concluding that she found it useful and carrying a positive potential. This illustrates how structural resources directly affect the nursing home employees’ daily work, constituting a challenge for successful implementation of the intervention. Here the need to prioritise one’s normal tasks at the nursing home and also to set up and facilitate activities such as the *Tell Stories for Life* intervention. This is likely to have affected the nursing home employees’ engagement and confidence in the intervention and its implementation.

Additionally, the many other activities at the nursing homes made it difficult for the nursing home employees to prioritise this particular intervention; and the residents interviewed explained it was difficult distinguishing this intervention from the other activities. Some of the nursing home employees had previous experiences or a personal interest in working with similar activities, which clearly acted as a motivational factor for them when implementing the intervention. However, this could not remove the barrier that lack of time and (too) many other activities posed in their daily work.

Although the intervention in this nursing home context appeared to carry little potential in terms of fulfilling its official aim, the nursing home employees articulated that nursing home residents and nursing home employees became more familiar with each other, that there was a perceived need to tell stories, and that it strengthened residents’ sense of identity. The nursing home employees explained in the interviews how they saw, participants ‘lighting up’ when given the chance to tell their stories, and that participants’ identity was strengthened through reliving past stories and memories that mattered to them. From the professional perspective, therefore, the intervention made sense to the nursing home residents and thereby to themselves, based on a perception that residents had a need for telling their stories and be listened to. This illustrated how the *Tell Stories for Life* intervention in spite of described limitations did seem to have other potential. By sharing life stories in smaller groups, participants ‘lit up’ and reminded themselves of who they were before moving into the nursing home. This way they might maintain links to what the *Tell Stories for Life* describes as their personal biography—that by telling their stories they might become new witnesses to each other’s lives. Therefore, although we did not observe that the nursing home residents established new social relations within the implementation period, that we were able to study, we argue that the intervention carries other positive implications for those involved. We point to a further exploration of whether the foundation for new relations in the future, might have been established through the course of this intervention as participants—residents and employees—get to know each other a little better during the short course of the intervention.

In line with this, it might be relevant to accept and acknowledge that it is challenging to measure success for a social intervention like this, and that there might be value in adapting the criteria of success to fit individual characteristics of those participating. For instance, in this case where the intervention is implemented among cognitively impaired nursing home residents, it might be worth considering the mere value of being listened to, sharing memories and thereby—at least for a while—escaping from narrative isolation and an identity as a nursing home resident.

Studies of similar interventions among older people in institutional settings have found a positive effect on sense of identity and cognitive performance [28], on quality of life and social engagement [24,31] and on wellbeing [26] which supports the assertion that the *Tell Stories for Life* intervention might yield potential benefits in a nursing home setting after all. Another study similarly remarks that a sense of purpose is important for nursing home residents’ quality of life, and that this can be established through a range of meaningful activities involving interactions with other people, hence moving away from the passive, isolated life that people may live in nursing homes [11]. Furthermore, more general work on the meaning of illness narratives supports our findings. Authors argue how sharing life stories can draw prospective maps as the individual tries to re-establish a sense of order from a discontinuity caused by major life events [45,46]: in this case, moving into a nursing home and life events preceding this major shift in life circumstances. For example, loss of spouse, relatives or friends can be causes of disruption in the older person’s life. Hence stories can be empowering, but at the same time fragile: shaped by social encounters, and situated and framed by the individual experiencing and telling the story [45,46].

A qualitative study by Lung and Liu supports the value of nursing home employees becoming more familiar with the nursing home residents during the intervention. They find, that a positive interaction not only improves the psychosocial wellbeing of the nursing home residents but also seems to translate into better cooperation and participation during the delivery of care which, as they suggest, might improve residents’ overall health and contribute to the nursing home employees job satisfaction [47]. However, as the authors emphasise, building close relationships through repeated reciprocal interaction takes time and hence, might not go hand in hand with temporary interventions. Furthermore, in line with Lung and Li, a cross-sectional study by Yang et al. points to the importance of a supportive psychosocial climate when creating good and person-centered interactions between residents and staff at nursing homes. Yang et al. also finds, like in this study, that it is important to include the perspectives of both residents and employees since these might differ substantially [48].

Although not focusing specifically on older people, a systematic review by Mann et al. presents an inspiring approach to categorise social interventions aimed at loneliness or related constructs among people with mental health problems [18]. Mann et al. propose to divide interventions into two types: firstly, those directly targeting loneliness and related concepts affiliated with social relationships; and secondly, interventions with a more indirect, broader approach, addressing wellbeing which might subsequently impact loneliness and related concepts [18]. This we argue, supports our argument of focusing on other potentials of the social intervention *Tell Stories for Life* such as strengthening sense of identity as this are likely to have a positive impact on wellbeing and subsequently maybe even the ability to establish new relationships with other people.

Based on this study of experiences with implementing the social intervention *Tell Stories for Life* in a nursing home context, we set out to explore processes and potentials. Overall, our results indicate that the *Tell Stories for Life* intervention did not meet the official aim of the intervention which was to strengthen and establish social relations among nursing home residents, and that it should be self-sustainable beyond the 10-week intervention period. However, interviews with the nursing home employees and participatory observations indicated some alternative positive effects of the intervention. Nursing home residents enjoyed participating and to have someone listening to their stories. Nursing home employees expressed confidence in the intervention as they saw a need for an intervention “like this”, and that it appeared to strengthen the identity of the nursing home participants and their relations to the nursing home employees. Moreover, in the view of the nursing home employees it also appeared to train residents’ rusty “talking skills” which carried benefits for future social encounters. Furthermore, our results point to the importance of adaptation to the local context and to the skills and perceptions of the nursing home employees implementing the intervention. Lastly, lack of structural resources was a key barrier for successful implementation at one of the nursing homes in this study.

We find it important to distinguish between assessing potentials if the aim is to fundamentally change social relationships through establishing new ones at old age and to make interventions sustainable over time versus if the aims are more modest, focusing on enhancing wellbeing through strengthening identity and counter a sense of narrative isolation. This might be particularly relevant considering a nursing home population group living in the autumn of their life. Also, the nursing home employees might take with them general and useful insights and experiences with life stories as a valuable tool in a nursing home setting. This way, in spite of doubts as to the interventions’ potential, we did see indications that the intervention made sense, that there is a need to share life-stories and that narratives might be a useful tool at nursing homes.

### 4.2. Limitations of the Study

In terms of sensitivity to the nursing home context, it was clear that nursing home residents participating in the study needed special consideration in terms of facilitating and aiding meaningful conversations both during the intervention sessions, and during the formal interviews. This observation underscores that a social intervention building on narratives and sharing life stories might not be the most suitable intervention for this particular group of older people. During the interviews with the nursing home residents, the first authors ought to introduce herself and have them ‘tune in’ to the social intervention. The first author did this the first time she met the nursing home resident and the second time, a short while after the intervention had finished. Still we found that some of the respondents had difficulties focusing on and remembering this social intervention. When encountering these difficulties in conducting the interviews, instead of the first author forcing her questions through, she adjusted and went for more open ended questions related to their social lives at the nursing homes in general.

Generally, doing fieldwork among cognitively impaired older people needs careful consideration. Older people living in nursing homes tend to have poorer health, lower functional and cognitive levels, and a weaker social network than older people in general [10,11,12,13]. This way, the nursing home residents’ vulnerabilities may be part of the reason why the accounts of the nursing home employees were substantially richer than those of the residents.

When recruiting residents to be part of the social intervention, the nursing home employees clearly reflected on and considered the residents’ ability to participate in such a narratively based social intervention; and secondly they carefully considered who to invite for the semi-structured interviews part of this study. This way, although not aiming at a purposive sampling as such, the selection of nursing home residents and employees included in this study, was not made by chance, but relied on the considerations of gatekeepers and their judgement of appropriateness and the residents’ abilities to participate. Initially, we envisioned that to achieve an acceptable level of saturation in the material, a larger sample size would be necessary. Inevitably, it might have strengthened the study if there were more informants, and hence more interviews with both residents and employees at the nursing home. However, we do believe that there is a satisfactory degree of data saturation despite the limited number of participants after all as the analysis of the 14 interviews uncovered several of the same themes relating to need, motivation, identity, and contextual barriers. Moreover, it has been argued that the generalisability of small-scale studies, particularly in nursing and other healthcare disciplines, should be that of a theoretical generalisation rather than an objective judgement based on tangible characteristics of the population being studied [49]. Finally, we argue that the findings do highlight important themes related to implementation of this social intervention that should be explored further.

We argue that future studies of social interventions among older people at nursing homes would benefit from more longer-term studies as this is a population for whom it might take a while to open up, share personal stories and thoughts and maybe, eventually, become friends. Furthermore, we recommend that future studies continue to apply a combination of interviews and participatory observations and if possible aim at including more nursing home residents and employees. It would also strengthen the insights if the implementation of the social intervention could be followed at several different settings in order to explore the importance of context further. This way it might be possible to gain a deeper and more varied insight into the perspectives of those implementing and participating in the social intervention. This way moving forward in order to achieve a better understanding of what makes a social intervention like the *Tell Stories for Life* successful.

## 5. Conclusions

In conclusion, we argue that although this specific social intervention, entitled *Tell Stories for Life*, does not seem to establish or strengthen social relations between nursing home residents, it might carry other potentials. However, if attention is not given to certain implementation processes and barriers—for example adaptation to local context such as nursing home residents’ cognitive level and time available for nursing home employees who are to implement the social intervention, it might not be perceived as successful.

Essentially, this means that the potential of social interventions using narratives at nursing homes should not be a priori rejected, but rather we need more research into the implementation, adaptation and effect of such interventions implemented specifically in nursing home settings. This is needed in order for us to move forward and inform the practical implementation of interventions such as *Tell Stories for Life* among nursing home residents and employees.

## Figures and Tables

**Table 1 ijerph-18-01808-t001:** Characteristics of the study population.

Study Participants	ID	Gender	Age	Years at Nursing Home
Nursing home residents				
	A	Female	3	<1
	K	Male	2	2
	F	Male	1	2
	E	Female	1	1
Nursing home employees				
	M	Female	-	<1
	T	Female	-	2
	ED	Female	-	3

## Data Availability

Data generated and analysed during the study will be stored and secured in a safe place. Restrictions apply to the availability of these data, which were used under license for the study, and so are not publicly available. Datasets used in this study will be available five years after publication.

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
