# Peer review of "Qualitative Study of the Implementation and Potential of a Social Intervention at Nursing Homes in Denmark"

_ijerph, 2021, doi:10.3390/ijerph18041808_

Round 1

Reviewer 1 Report

Thank you for the opportunity to review this article.

The research is interesting; I've some issue Authors should address:

references are not updated; Authors should review the literature to check if new evidence has been produced in this regard. It is advisable to re-access the indicated internet pages (line 705-707; 717-719; 751-752; 756-757) and assess whether the data reported are consistent as two years after the last access.

Some typos and repetitions due to copy and paste in the paper, especially in the final stages of the item, such as line 594 "sThis", line 598, 599, 604, 629 repeating or absent spaces. We recommend that proofreading perform a review.

Data were collected in a sample of 4 patients and 3 nurses. Being a qualitative research I wonder: the small number of people involved for data analysis would seem to be a limiting factor, did you get the same data saturation? Or has the limited number been maintained according to the situation indicated in the article? Authors should better explain their process.

The conclusion paragraph shows that the use of the narrative technique "Tell Stories for life", is positive about implementing the social impact and reducing the isolation of the elderly in nursing homes, but from the results obtained, line n239/244, 247/250, 251/253, 258/260 and 416/420, it turns out that the technique did not bring the desired results as the patients did not increase the relationship with the other elderly.

As the authors describe, the technique has been enthusiastically collected. The participants (the patients) are in favour of repeating it, identifying it as a recreational activity, but only in the presence of the guide nurse (line: 239-244; 254-256), and the nurses complain of lack of time to propose it not only as a recreational activity (line: 381-382).

The limit proposed by Nurse M, line 326-332, is part of the social obstacles, it would be to be identified as a possible limit of the study that makes it difficult to share among the people living in the nursing home. I think the same for the ED comment, line 361-368: the project in focus group is well described. But it seems too much structured. Authors can describe their choice to set the focus this way?

Author Response

Reviewer 1

References are not updated; Authors should review the literature to check if new evidence has been produced in this regard. It is advisable to re-access the indicated internet pages (line 705-707; 717-719; 751-752; 756-757) and assess whether the data reported are consistent as two years after the last access.

Thank you very much for this helpful comment. We have reviewed and checked the referenced web pages (ref. nr 10, 16, 32 and 35) and updated the dates accordingly. Furthermore, we have added a few new references in the Background section (please see lines 54-67 and lines 81-97).

Some typos and repetitions due to copy and paste in the paper, especially in the final stages of the item, such as line 594 "sThis", line 598, 599, 604, 629 repeating or absent spaces. We recommend that proofreading perform a review.

Thank you for pointing our attention to these typos. They have all been corrected.

Data were collected in a sample of 4 patients and 3 nurses. Being a qualitative research I wonder: the small number of people involved for data analysis would seem to be a limiting factor, did you get the same data saturation? Or has the limited number been maintained according to the situation indicated in the article? Authors should better explain their process.

Thank you very much for this comment. We acknowledge that the data material derives from relatively few interviewees whom have all been interviewed twice (total nr of interviews=14). We have sought to elaborate on recruitment and data saturation in the Discussion section (please see lines 708-730).

The conclusion paragraph shows that the use of the narrative technique "Tell Stories for life", is positive about implementing the social impact and reducing the isolation of the elderly in nursing homes, but from the results obtained, line n239/244, 247/250, 251/253, 258/260 and 416/420, it turns out that the technique did not bring the desired results as the patients did not increase the relationship with the other elderly.

Thank you for this comment. The conclusion has been shortened somewhat, following advice from reviewer 3, and the conclusion now reads as follows:

In conclusion, we argue that although this specific social intervention, entitled Tell Stories for Life, does not seem to establish or strengthen social relations between nursing home residents, it might carry other potentials. However, if attention is not given to certain implementation processes and barriers—for example adaptation to local context such as nursing home residents’ cognitive level and possible barriers, it might not be perceived as successful. Essentially, this means that the potential of social interventions using narratives at nursing homes should not be a priori rejected, but rather we need more research into the implementation, adaptation and effect of such interventions implemented specifically in nursing home settings. This is needed in order for us to move forward and inform the practical implementation of interventions such as Tell Stories for Life among nursing home residents and employees.” (please see lines 757-788).

This should hopefully underline that this specific intervention does not seem to establish social relations and/or reduce social isolation among elderly in nursing home. However, we argue that primarily through the lens of the nursing home employees, we observe other potentials such as strengthening the individual sense of identity among residents and the connectedness between resident and employee.

As the authors describe, the technique has been enthusiastically collected.The participants (the patients) are in favour of repeating it, identifying it as a recreational activity, but only in the presence of the guide nurse (line: 239-244; 254-256), and the nurses complain of lack of time to propose it not only as a recreational activity (line: 381-382).

Thank you very much for your positive feedback.

The limit proposed by Nurse M, line 326-332, is part of the social obstacles, it would be to be identified as a possible limit of the study that makes it difficult to share among the people living in the nursing home. I think the same for the ED comment, line 361-368: 

We agree that there are important limitations to consider related to the implementation of social interventions specifically at nursing homes. This we have tried to reflect upon in the Discussion section (please see lines 684-707 and again in lines 740-754).

The project in focus group is well described. But it seems too much structured. Authors can describe their choice to set the focus this way?

We have sought to describe the flow in the group sessions as transparent as possible although there were variations in the details of how the sessions unfolded (please see lines 336-355, 395-430).

Reviewer 2 Report

You shared "In summary, our results indicate that the Tell Stories for Life 416 intervention did not meet the official aim of the intervention which 417 was to to strengthen and establish social relations among nursing 418 home residents, and that it should be self-sustainable beyond the ten 419 week intervention period." I would add that since this is an elderly population and it takes longer for elderly to make friends and feel safe in opening up in a group setting, future studies should include elderly that are in these homes for at least 2 years rather than the one year that you used in this study.  I recommend adding this to the recommendations of future studies.

Author Response

Reviewer 2

You shared "In summary, our results indicate that the Tell Stories for Life 416 intervention did not meet the official aim of the intervention which 417 was to strengthen and establish social relations among nursing 418 home residents, and that it should be self-sustainable beyond the ten 419 week intervention period." I would add that since this is an elderly population and it takes longer for elderly to make friends and feel safe in opening up in a group setting, future studies should include elderly that are in these homes for at least 2 years rather than the one year that you used in this study.  I recommend adding this to the recommendations of future studies.

Thank you for this very concrete and helpful comment. We have briefly added a paragraph relating to this issue to the Results section (please see lines 388-394) and the Discussion section (please see lines 740-744).

Reviewer 3 Report

The study focuses on important aspect related to nursing work in the nursing homes. It is based on the implementation of a social intervention entitled Tell Stories for Life among elderly in nursing home. The manuscript is well prepared, there are several issues however, which may help improve the quality of this paper:

  • when describing setting and recruitment to the study, please consider to add flow chart showing step by step the complexity of the process. Heving only description it is difficult to follow.
  • in results section, lines 231-232, when writing about anonimity and names of participants - please, move this sentence to ethical considerations.
  • in results section, lines 416-434, this extract should be moved to the discussion, as include kind of interpretation of your results. Additionally, this is repeated several times, so you may revise this issue to avoid repetition (e.g. et the beginning and at the end of results).
  • the section 'methodological reflections' - should be titled 'Limitation of the study'.
  • in conclusions, please do not repeat again your results. You can make conclusions shorter focusing on implications coming from your study.

Author Response

Reviewer 3

The study focuses on important aspect related to nursing work in the nursing homes. It is based on the implementation of a social intervention entitled Tell Stories for Life among elderly in nursing home. The manuscript is well prepared, there are several issues however, which may help improve the quality of this paper:

When describing setting and recruitment to the study, please consider to add flow chart showing step by step the complexity of the process. Heving only description it is difficult to follow.

Thank you for this comment. In keeping with the qualitative approach, we believe a more narrative description of the study, including the intervention, recruitment of participants and data collection, is more suitable.

In results section, lines 231-232, when writing about anonimity and names of participants - please, move this sentence to ethical considerations.

Thank you for this comment. We have moved the sentence to ethical considerations (please see lines 263-265).

In results section, lines 416-434, this extract should be moved to the discussion, as include kind of interpretation of your results. Additionally, this is repeated several times, so you may revise this issue to avoid repetition (e.g. et the beginning and at the end of results).

Thank you for this comment as well. The paragraph has been moved to the discussion section where I find it to be more suitable (please see lines 651-669).

The section 'methodological reflections' - should be titled 'Limitation of the study'.

Thank you for this comment. The title of the section has been changed accordingly (please see line 683).

In conclusions, please do not repeat again your results. You can make conclusions shorter focusing on implications coming from your study.

Thank you very much for this comment. The conclusion has been revised and considerably shortened (please see lines 771-788).

Reviewer 4 Report

Text issues

239 277 599 (review spaces)
594 tipping error

method

the criteria of selecting the sample and the individual characterists needed could be more explicit.

results
so that nursing home employees might diverge from running 353 it according to the described model
If should be show more direct the edidence of the intervention in the individuals and in the group.
I suggest exploring more the impact of the dedicated time of this interventions on professionals. Maybe exploring more the pros and cons.
The results conclude that is also important to communicate that some interventions dont have the expected result. congratulates for that.

Discussion
The issue about professionals time spend is pointed and give a first impressionon of the related itens.

Author Response

Reviewer 4

239 277 599 (review spaces)

594 tipping error

Thank you for noting these typos. They have been corrected in the revised version.

Method

The criteria of selecting the sample and the individual characterists needed could be more explicit.

Thank you very much for this comment. Reflections on recruitment and saturation have been added (please see lines 708-730).

results
so that nursing home employees might diverge from running 353 it according to the described model
If should be show more direct the edidence of the intervention in the individuals and in the group.
I suggest exploring more the impact of the dedicated time of this interventions on professionals. Maybe exploring more the pros and cons. The results conclude that is also important to communicate that some interventions dont have the expected result. Congratulates for that.

Thank you for these valuable reflections. Given the depth and scope of the data generated in this study, we are unfortunately not able to dichotomise results according to individual level and group levels respectively. The same aspects prevent a further exploration of the importance of time on the side of professionals. We will however, carefully consider including this in future studies as both dimensions are interesting and under-explored.

Discussion
The issue about professionals time spend is pointed and give a first impressionon of the related itens.

Thank you. Please see our response to the previous comment.

Round 2

Reviewer 1 Report

Thank you for the revised version of the paper.
This research is exciting and provides a new way to interact with older adults in nursing homes. It deserves more investigations to show how it could be implemented in the routine for patients and carers.
all suggestion provided have been fulfilled

Reviewer 3 Report

Authors provided changes in a manuscript following recommendations given